# Longitudinal transcriptional changes reveal genes from the natural killer cell-mediated cytotoxicity pathway as critical players underlying COVID-19 progression

Matias A Medina[1], Francisco Fuentes-Villalobos[2], Claudio Quevedo[3], Felipe Aguilera[3], Raul Riquelme[1,4], Maria Luisa Rioseco[1,4], Sebastian Barria[1,4], Yazmin Pinos[5], Mario Calvo[6], Ian Burbulis[1], Camila Kossack[1], Raymond A Alvarez[7], Jose Luis Garrido[1]*, Maria Ines Barria[1]*

[1]Facultad de Medicina y Ciencia, Universidad San Sebastián, Puerto Montt, Chile; [2]Departamento de Microbiología, Facultad de Ciencias Biológicas, Universidad de Concepción, Concepción, Chile; [3]Departamento de Bioquímica y Biología Molecular, Facultad de Ciencias Biológicas, Universidad de Concepción, Concepción, Chile; [4]Hospital Dr. Eduardo Schütz Schroeder, Puerto Montt, Chile; [5]Hospital Base San José, Osorno, Chile; [6]Instituto de Medicina, Facultad de Medicina, Universidad Austral, Valdivia, Chile; [7]Division of Infectious Diseases, Department of Medicine, Immunology Institute, Icahn School of Medicine at Mount Sinai, New York, United States

*For correspondence:
jlgarri@gmail.com (JLG);
maria.barriac@uss.cl (MIB)

**Competing interest:** The authors declare that no competing interests exist.

## eLife assessment

This **valuable** paper compares blood gene signature responses between small cohorts of individuals with mild and severe COVID-19. The authors provide **solid** evidence for distinct transcriptional profiles during early COVID-19 infections that may be predictive of severity, within the limitations of studying human patients displaying heterogeneity in infection timelines and limited cohort size.

**Abstract** Patients present a wide range of clinical severities in response severe acute respiratory syndrome coronavirus 2 infection, but the underlying molecular and cellular reasons why clinical outcomes vary so greatly within the population remains unknown. Here, we report that negative clinical outcomes in severely ill patients were associated with divergent RNA transcriptome profiles in peripheral immune cells compared with mild cases during the first weeks after disease onset. Protein–protein interaction analysis indicated that early-responding cytotoxic natural killer cells were associated with an effective clearance of the virus and a less severe outcome. This innate immune response was associated with the activation of select cytokine–cytokine receptor pathways and robust Th1/Th2 cell differentiation profiles. In contrast, severely ill patients exhibited a dysregulation between innate and adaptive responses affiliated with divergent Th1/Th2 profiles and negative outcomes. This knowledge forms the basis of clinical triage that may be used to preemptively detect high-risk patients before life-threatening outcomes ensue.

## Introduction

The severe acute respiratory syndrome coronavirus 2 (SARS-CoV-2) promotes several dysfunctions in human immune responses while triggering a broad spectrum of clinical presentations that range from asymptomatic infection to a mild, moderate, or sometimes lethal severe symptomatology (*Ge et al., 2020*; *The, 2020*; *Wu and McGoogan, 2020*). Convalescent patients report prolonged COVID-19 symptoms beyond the time course for typical cold and flu events, which highlights the possibility of long-term tissue damage generated by this virus (*Ladds et al., 2020*; *Nalbandian et al., 2021*; *Ryan et al., 2022*; *Subramanian et al., 2022*). Similarly as other respiratory viruses, it is known that SARS-CoV-2 triggers an immune response involving the recruitment, activation, and differentiation of innate and adaptive immune cells (*Newton et al., 2016*; *Shen et al., 2023*; *Wauters et al., 2021*). For mildly ill patients, these coordinated immunological efforts resolve infection but for unknown reasons the virus evades these responses in severely ill patients to produce life-threatening COVID-19 (*Park, 2020*; *Rashid et al., 2022*; *Sun et al., 2022*; *Thorne et al., 2022*). The genetic background and physiological health of individual patients certainly plays a major role in the clinical presentation of COVID-19 but the exact mechanisms of how the virus evades innate and adaptive responses is not known, or why there is such great variability in severity of clinical presentation among patients. This information is critical for developing new diagnostics that detect patients who will eventually progress to severe COVID-19 before respiratory failure ensues, and furthermore, provide host and virus targets to engineer effective treatments (*Li et al., 2021b*; *Samadizadeh et al., 2021*).

Biomarkers linked to COVID-19 severity hold promise for detecting patients that will eventually develop severe COVID-19 (*Janssen et al., 2021*; *The, 2020*). In this context, blood-derived cues were associated with severe COVID-19, including an imbalance in immune cell populations that included neutrophil abundance, lymphopenia, myeloid dysfunction, and T cell activation/exhaustion (*Ahern et al., 2022*; *Chen and John Wherry, 2020*; *Mann et al., 2020*; *Wauters et al., 2021*). The differential expression of select chemokines and their receptors (*Khalil et al., 2021*) with associated cytokine storm drove monocyte and megakaryocyte dysfunction in severely ill patients (*Ren et al., 2021*). Comprehensive knowledge of host immune responses against SARS-CoV-2 is still limited but these divergent cell profiles implicated cell-to-cell signaling events occurring between the innate and adaptive cell compartments as critical for the progression of severe COVID-19 (*Daamen et al., 2021*; *Rabaan et al., 2023*; *Wang et al., 2022*).

One approach to identifying changes in immune responses affiliated with severe COVID-19 is to monitor autocrine, paracrine, and endocrine signaling in individual patients over time. Temporal events associated with each type of signaling is obviously difficult to disentangle from measuring the activities of circulating peripheral cells alone because there are distinct events happening in localized microenvironments, for example, the spleen and lymph nodes. A complementary tactic to access information about these events is to monitor gene expression for the synthesis of chemokines and cell-associated receptors as a proxy of biochemical events happening in distinct immune effector cells. Based on current knowledge, we hypothesized that critical events occurring at the earliest stages of infection necessary for effective viral clearance are either perturbed or disrupted so as to promote cytokine storm and other pathologies associated with severe outcomes. We predicted these pathological immune events may be observable by measuring changes gene expression reflecting activities in distinct effector cells during the first weeks of infection (*Ahern et al., 2022*; *Bernardes et al., 2020*; *Notarbartolo et al., 2021*; *Xiong et al., 2020*; *Zheng et al., 2020*). However, these types of experiments require careful design because the type and quantity of all immune responses are dynamic during infections and comparing poorly matched peripheral blood mononuclear cells (PBMCs) may confound identification of bonafide immune dysregulation evident between patients (*Bernardes et al., 2020*; *Notarbartolo et al., 2021*; *Zheng et al., 2020*).

Here, we designed a longitudinal comparison between mild and severe patients, choosing the appropriate samples according to the clinical progression and the unbiased gene expression profile to study how changes in gene expression in distinct immune effector cells changed during the earliest time points since peak of symptoms and during progression of clinical disease. We repeatedly measured whole-transcriptome profiles of PBMCs from the same cohort of mildly and severely ill patients to identify molecular pathways that were enriched during the clinical trajectory of COVID-19 over time. Briefly, to gain more insights into our findings and get more comprehensive knowledge of the phenomenon, we used a pairwise comparison of gene expression, gene set enrichment, and

weight-correlated gene network analyses. By doing so, we identified pathways of genes involved with the natural killer (NK) cell cytotoxicity enriched in mild patients when compared to severe. Besides focusing on a particular molecular pathway, we investigated the interactions to better comprehend the underlying phenomena of a successful immune response, contributing to an integrated point of view throughout the transcriptomic analyses of functional pathways to mitigate potential biases attributed to focusing the study on a single pathway. In this regard, we revealed that the NK signaling pathway was intricately related to other transcriptional circuits, such as those governing Th1/Th2 cell differentiation and cytokine–cytokine receptor signaling pathways. These interactions highlight the importance of these pathways as bridges between the innate and adaptive immune responses throughout the disease, implying that the innate NK signaling pathway (cell cytotoxic activity) is beneficial, and possibly a critical activity required to effectively eradicate coronavirus. We also observed that an adaptive immune response including early cell-mediated immunity was significant in lowering disease severity. The link between the primary innate NK cell activity and the transcriptional priming of adaptive Th1 and Th2 cell responses appears to be more robust in mild patients than in severe. This work provides clear guidance to develop better medical practices and prevention tactics against SARS-CoV-2 and other related infectious respiratory virus (*Haitao et al., 2020*; *Ponti et al., 2020*; *Zhang and Guo, 2020*).

## Results
### Clinical features and temporal gene expression patterns in SARS-CoV-2 infected patients

A total of 22 peripheral blood samples were obtained from eight COVID-19 patients. These samplings following a longitudinal schedule complemented with two samples from healthy donors. All patients were recruited after an average period of 5 days after symptoms onset (*Figure 1A*). Some samples were taken from patients at the Hospital of Osorno and the Hospital of Puerto Montt, which are cities located in the region Los Lagos. The remaining samples were collected from patients at the Hospital Base and Clínica Alemana in Valdivia, a city located in the Region Los Ríos. All infections occurred between November 2020 and May 2021 (*Table 1*).

A crucial issue with longitudinal studies is defining an appropriate sampling schedule that provides a reasonable comparison between patients during the time course of naturally occurred infections. To align the comparability and consistency of data measured between patients, we designed a protocol consisting of three donations per patient to monitor events occurring during both acute infection and the recovery phase. We collected peripheral blood samples on days 0, 7, and 28 (D0, D7, and D28) during the peaks of symptoms (*Figure 1B*).

Clinical features of COVID-19 patients with mild and severe symptoms were determined by medical personnel at the hospitals mentioned above and used to describe their disease trajectories over time using the WHO ordinal scale (*Ahern et al., 2022*, *Figure 1B*). In contrast to mild patients, all four severe patients experienced symptoms such as fever, cough, headache, chills, diarrhea, myalgia, and dyspnea (*Table 1*). These patients received mechanical ventilation on sampling D0. In general, symptoms from severe and mild patients diminished gradually up to D28 after recruitment, with the exception of one mild and another severe patient who still experienced mild symptoms.

Mild and severely ill patients displayed different transcriptional programs at the beginning of disease onset. To determine the gene expression profiles of each patient over the time of disease progression, we developed an RNA-seq approach that takes advantage of the longitudinal sampling scheme. By using all expressed genes, we performed a principal component analysis to unbiasedly compare the transcriptional signatures of each patient and two healthy donors. All peripheral blood samples from severe patients on D0 (represented by red circles) were widely dispersed over the left of principal component 1 (PC1) (*Figure 1C*), in contrast with mild patients on D0 (represented by blue circles), suggesting that both groups of patients displayed different transcriptional programs at the beginning of the disease.

The transcriptional profiles of severely ill patients changed during the recovery phase to be consistent with that observed in mildly ill patients. Gradually, along with disease progression and medical treatments, samples from severely ill patients shifted to the right of PC1 (D7 and D28). Interestingly, on D28, when the majority of patients had recovered, samples from severely ill patients were pooled

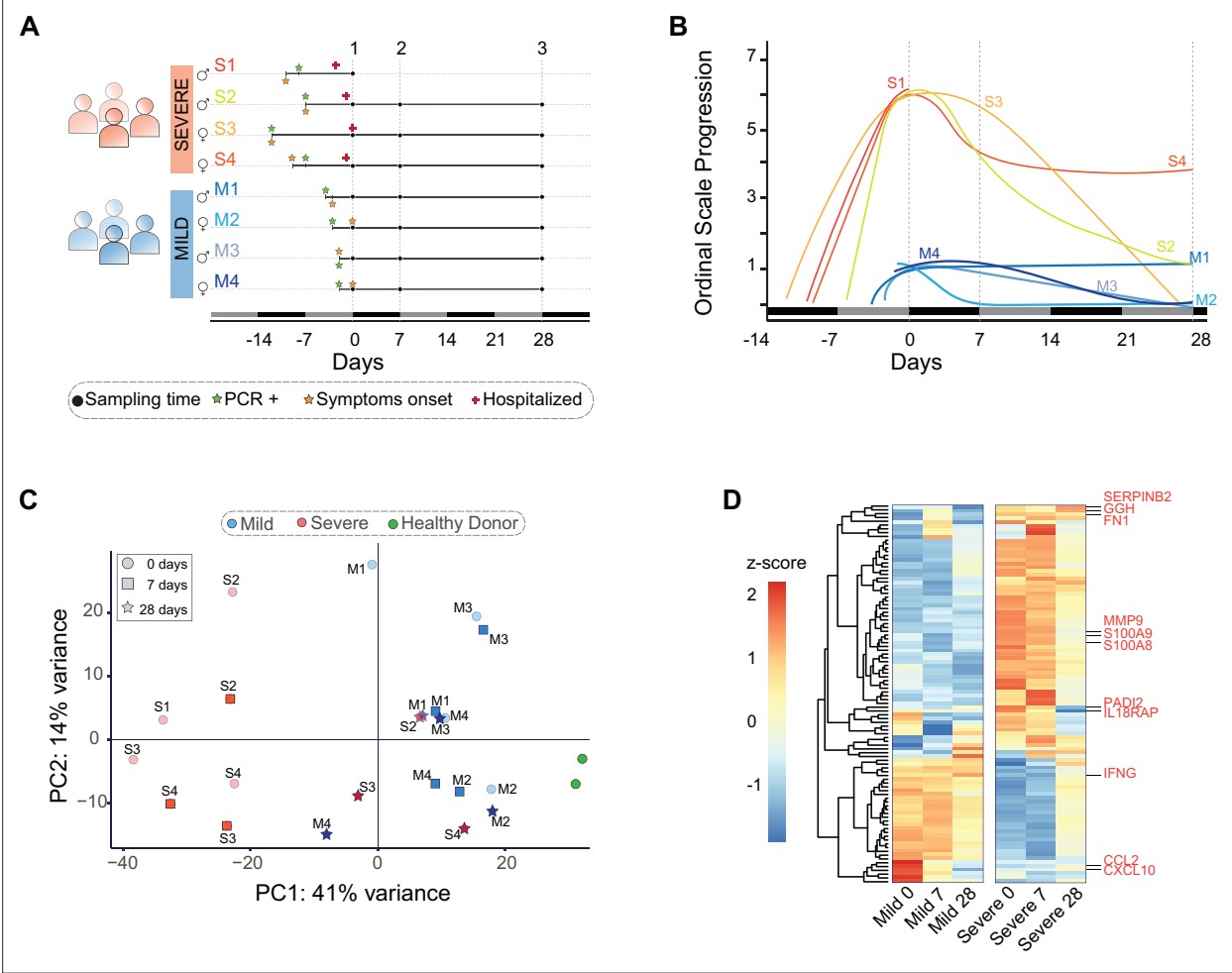

**Figure 1.** Clinical profile and gene expression patterns for mild and severe COVID-19 patients during 28 days. (**A**) Longitudinal sampling schedule for severe and mild COVID-19 donors of peripheral blood (22 samples in total from 8 donors). Three sampling times (black dots) are displayed with respect to the recruitment day (D0). In addition, panel A shows the diagnosis day with positive PCR (green star), symptoms onset day (orange star), and hospitalization day (red cross). (**B**) The COVID-19 progression according to the WHO ordinal scale describes the temporal disease severity for each donor. Severe patients (S1–S4) are displayed in lines with colors scaling from green to red, while mild patients (M1–M4) are shown in blue line colors. The *X*-axis represents the days relative to the recruitment day. The vertical dot lines indicate the three sampling times used in this study (days 0, 7, and 28). (**C**) Principal component analysis plot based on gene expression profiles for mild (blue) and severe (red) COVID-19 patients and grouped by their sampling time (0, 7, and 28 days after recruitment). In addition, peripheral blood samples from two healthy donors are shown in green. PC = principal component. (**D**) Heatmap of the 100 most significant differentially expressed genes related to the COVID-19 disease progression. At the bottom, each column corresponds to the sampling points (0, 7, and 28 days since recruitment) of mild and severe patients. Genes are displayed as horizontal rows and are clustered by the similarity of expression profiles, represented by the dendrogram to the left of the heatmap. Red indicates higher expression, while blue means lower expression represented by the *z*-score of normalized read counts. Some COVID-19 severity-associated genes previously reported are indicated to the right of the heatmap in red.

The online version of this article includes the following figure supplement(s) for figure 1:

**Figure supplement 1.** Heatmap of temporally and differentially expressed genes over the course of COVID-19 progression.

**Figure supplement 2.** Volcano plot depicting pairwise gene expression comparisons for detected differentially expressed genes (DEGs) between mild and severe COVID-19 patients at D0, D7, and D28 (**A, B, and C**, respectively).

together with those mild patients who had already recovered. These observations indicated that despite the transcriptional profiles being closer to that of mild patients at D28 as compared to D0, severely ill patients still exhibited higher variability between themselves and controls (*Figure 1C*). In contrast, every mild COVID-19 patient was separated from the severe group on D0 and D7. Notably, only one mild COVID-19 patient (M1) clustered with severe patients at D0. This donor showed a broader set of symptoms over time when compared with the rest of mild patients (*Figure 1B*). This

**Table 1.** Clinical characteristics of the cohort.

| Clinical characteristics | Mild | Severe | All |
|---|---|---|---|
| Sex, *n* (female/male) | (2/2) | (2/2) | (4/4) |
| Total, *n* | 4 | 4 | 8 |
| Median age, years ± SD | 39.0 ± 3.9 | 46.7 ± 8 | 42.9 ± 7 |
| Days from onset of symptoms to recruitment, median ± SD | 1.2 ± 1.3 | 10.0 ± 1.8 | 5 ± 4.7 |
| Days from COVID-19 diagnosis to recruitment, median ± SD | 3.0 ± 0.7 | 8.5 ± 2.1 | 6 ± 3.4 |
| | | | |
| Symptoms, *n* (%) | | | |
| Fever | 1 (25%) | 3 (75%) | 4 (50%) |
| Chills | 1 (25%) | 2 (50%) | 3 (37.5%) |
| Fever feeling | 1 (25%) | 2 (50%) | 3 (37.5%) |
| Odynophagia | 2 (50%) | 1 (25%) | 3 (37.5%) |
| Cough | 1 (25%) | 4 (100%) | 5 (62.5%) |
| Expectoration | - | - | - |
| Dyspnea | - | 4 (100%) | 4 (50%) |
| Thoracic pain | 1 (25%) | 1 (25%) | 2 (25%) |
| Diarrhea | 1 (25%) | 2 (50%) | 3 (37.5%) |
| Anosmia | 1 (25%) | - | 1 (12.5%) |
| Ageusia | 1 (25%) | - | 1 (12.5%) |
| Myalgia | 1 (25%) | 2 (50%) | 3 (37.5%) |
| Headache | 2 (50%) | 1 (25%) | 3 (37.5%) |
| | | | |
| Treatment | | | |
| Hospitalization, *n* (%) | - | 4 (100%) | 4 (50%) |
| Intubation, *n* (%) | - | 4 (100%) | 4 (50%) |
| Mechanical ventilation, *n* (%) | - | 4 (100%) | 4 (50%) |
| | | | |
| Samples, *n* | 12 | 10 | 22 |

evidence indicated a dissimilar transcriptional response in that specific patient at the onset of disease. Over time, and after medical treatments, the transcriptional program of this patient shifted to be consistent with the other mild patients (*Figure 1C*).

The timing of COVID-19-related gene expression differed between mild and severely ill patients. We focused on the temporal variation of gene expression to identify differentially expressed genes (DEGs) associated with COVID-19 progression. We found statistically significant differences in the timing of differential gene expression between mild and severely ill individuals (*Figure 1D* and figure supplement 1). We observed that severe patients displayed a transcriptional response completely different from that of mild patients at the sequential time points of D0 and D7 (*Figure 1D*). Previous longitudinal studies identified molecular markers associated with severe COVID-19 (*Bernardes et al., 2020*; *Notarbartolo et al., 2021*; *Zheng et al., 2020*). We detected these same molecular markers in our severely ill cohort (*Figure 1D*). The expression profiles of those genes varied significantly between mild and severe patients. For instance, the expression of MMP9 metalloproteinase (*Zheng et al., 2020*), S100A8/A9 alarmins (*Bernardes et al., 2020*), PADI2 (*Notarbartolo et al., 2021*), and IL18Rap peptidyl-arginine deiminases (*Masood et al., 2021*; *Schultze and Aschenbrenner, 2021*)

were higher in severe patients on D0 than mild or control patients (*Figure 1D*). In addition, we found that IFNG, CCL2, and CXCL10 cytokines, which were previously described as molecular markers in severely ill patients (*Sette and Crotty, 2021*; *Vabret et al., 2020*), displayed low expression in our severe COVID-19 patients in comparison with mildly ill patients during the progression of disease (*Figure 1D*).

## The immune response of mild and severe patients is activated differentially during the acute phase of the COVID-19 infection

Most of the variations observed in the gene expression profiles of mild and severely ill patients occurred during the acute phase of disease. We performed pairwise gene expression comparisons between mild and severe patients and found DEGs mainly on D0 and D7. On D0, we found a total of 812 DEGs including 298 upregulated and 514 downregulated genes (*Figure 1—figure supplement 2*). On D7, the number of DEGs was similar to D0, with 319 genes showing higher expression and 563 genes with lower expression. We found no differential gene expression between mild and severe patients at D28, supporting the interpretation that most imbalances in the gene expression profiles in the PBMCs of severely ill patients leveled out by D28 (*Figure 1—figure supplement 2*).

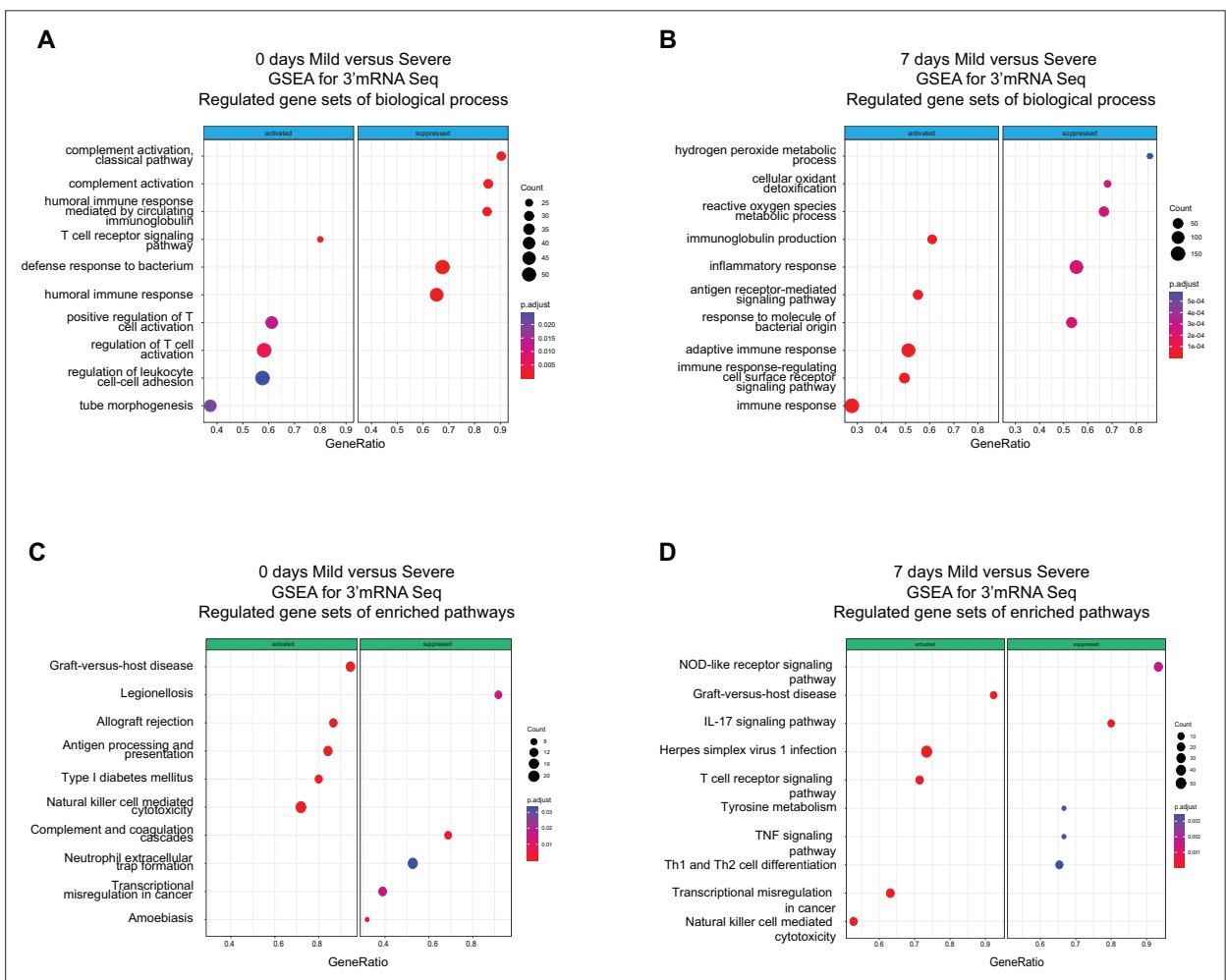

**Figure 2.** Gene set enrichment analysis (GSEA) shows biological processes and enriched pathways associated to innate and adaptive immune response after severe acute respiratory syndrome coronavirus 2 (SARS-CoV-2) infection. (**A, B**) Bubble plots show the biological processes (BP) of enriched genes for differentially expressed genes (DEGs) between mild and severe for D0 and D7 after recruitment, respectively. (**C, D**) Bubble plots show the Kyoto encyclopedia of gene and genomes (KEGG) enriched pathways of enriched genes for DEGs between mild and severe for D0 and D7, respectively. The *Count* (black circles) represents the number of genes included on each set. *Generatio* is the ratio between number of genes found in the set and total genes of set. The scale-color bar indicates the p-adjustment of each BP or KEGG pathway.

Functional pathways involved with humoral immunity were enriched in severely ill patients during the acute phase compared to pathways involved with cell-mediated immunity in mild patients. The above results provided only a course overview of the transcriptional responses during COVID-19 progression. We expanded our focus to detect molecular mechanisms and pathways involved in the immune responses of all patients by linking functional pathways to deferentially expressed genes (DEGs) detected between severely ill, mildly ill and control patients. We used a twofold change in gene expression level as a threshold to identify DEGs between mild and severe patients on D0 and D7. We found upregulated expression for genes involved in biological processes that included the T receptor signaling pathway, positive regulation of T cell activation, and regulation of leukocyte cell adhesion in mild COVID-19 patients at D0 (*Figure 2A*). We observed genes involved with immunoglobulin production, antigen receptor-mediated signaling pathway, and adaptive immune response were upregulated at D7 (*Figure 2B*). In contrast, we observed enrichment of gene expression in pathways involved with complement activation, humoral immune response mediated by circulated immunoglobulin, and defense response to bacterium on D0 in severe COVID-19 patients. Furthermore, DEGs in functional pathways mediating hydrogen peroxide metabolic processes, cellular oxidant detoxification, and reactive oxygen species were enriched on D7 of infection in this group (*Figure 2A, B*). Biological pathways consistent with a robust lymphocyte cellular immune response were enriched on D0 in mild patients. This functional profile is distinctly different to the antibody/complement-dependent humoral immune responses observed in severely ill individuals at the same time point (*Figure 2A*). Nonetheless, differential expression of genes associated with immunoglobulin function were mainly enriched in mild patients at D7 (*Figure 2B*), while severe patients showed enrichment for genes related to inflammation, reactive oxygen species, and responses against bacteria at that time of infection (*Figure 2B*).

In addition to enriched biological processes, we also focused on Kyoto encyclopedia of gene and genomes (KEGG) pathway enrichment among DEGs at D0 and D7 after COVID-19 infection. On D0, mild and severe patients showed considerable differences in terms of the innate response, with the NK-mediated cytotoxicity pathway enriched in mild-infected patients, while neutrophil extracellular trap formation was enriched in severe ones (*Figure 2C*). Furthermore, DEGs associated with the antigen processing and presentation pathways are enriched in mild COVID-19 patients, in contrast with the enrichment of complement and coagulation cascade pathways in severely ill patients (*Figure 2C*). On the other hand, on D7 of the COVID-19 infections, NK cell-mediated cytotoxicity is one of the main enriched pathways in the mild-infected group, whereas IL-17 signaling is the most significant pathway in the severe-infected group (*Figure 2D*). This finding is remarkable because besides COVID19, IL-17 is affiliated with other clinical pathophysiologies in which a dysregulation between innate and adaptive immune responses such as myocarditis and lupus (*Lee et al., 2019*; *Rangachari et al., 2006*; *Sadeghi et al., 2021*).

Taken together, we show that there are distinct transcriptional responses along the COVID-19 progression, which suggest that immune responses to SARS-CoV-2 infection occur differently in individuals; thus, there might exist a differential imprinting associated with the severity of the COVID-19 infection.

## Higher expression of NK cell hub-genes is a core event of acute phase that distinguishes mild from severe symptoms in COVID-19 individuals

Given that our findings pointed out changes in the immune response after SARS-CoV-2 infection of the patients cataloged as mild and severely ill, we decided to uncover molecular pathways that might be responsible of the differences observed between patient groups during COVID-19 progression. To do so, we first identified genes that were differentially expressed between severity groups, and second, we chose only those that also showed changes in their trajectories across sampling times. In doing so, we found 828 genes that exhibited temporal differences in expression level during disease progression. Then using the Enrichr platform, we discovered additional biological processes and KEGG pathways that were differentially enriched during the COVID-19 progression in mild and severe patients (*Figure 3*). For instance, mild-infected patients exhibited expression of genes involved in kinase activity, enzyme-linked receptor activity, and apoptotic process not only at D0 (acute phase) but also at D7 (middle phase) (*Figure 3A, C*). In contrast, severely ill patients exhibited high level expression of genes involved in neutrophil activity. This observation was the most notorious outcome

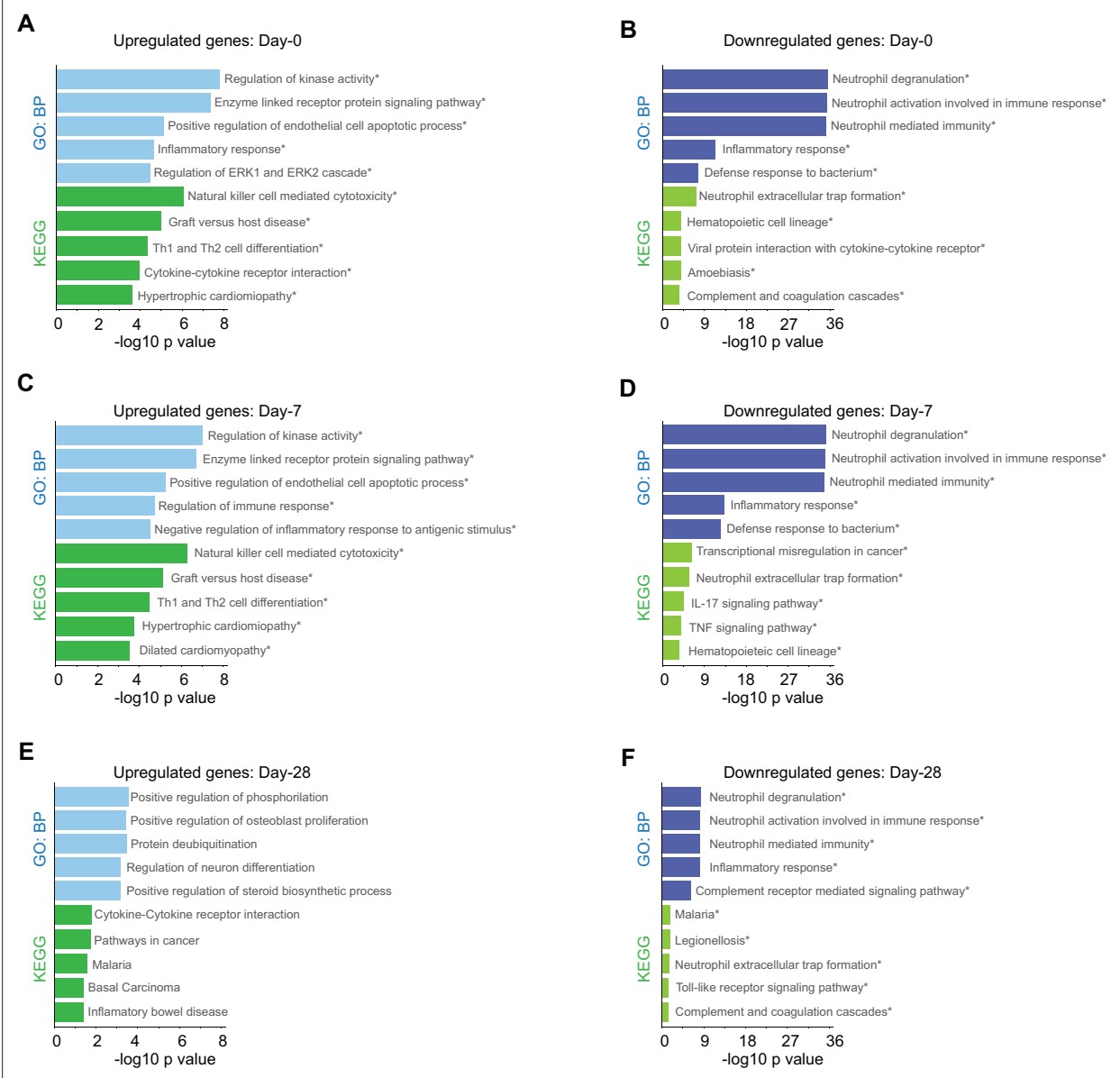

**Figure 3.** Biological processes and Kyoto encyclopedia of gene and genomes (KEGG) pathway for genes with differential expression levels over time in mild versus severe patients. Bar plots show gene ontology analysis for biological processes in light-blue/dark-blue and KEGG pathway in light-green/dark-green, using upregulated genes (**A, C, E**) and downregulated genes (**B, D, F**). The size of each bar is according to its −log10 p-value, and name pathways with asterisk indicates a *q*-value ≤0.05.

The online version of this article includes the following source data for figure 3:

**Source data 1.** Enriched pathways from genes upregulated in severe COVID-19 patients by longitudinal analysis for *Figure 3A, C, E*.

**Source data 2.** Enriched pathways from genes upregulated in severe COVID-19 patients by longitudinal analysis for *Figure 3B, D, F*.

elicited by SARS-CoV-2 during acute COVID-19 in this group (*Figure 3B, D, F*). We observed that NK cell cytotoxicity was the most enriched pathway among the temporal and differential expressed genes in mildly ill patients during the acute phase (*Figure 3B, C*). Among these enriched genes, we found abundant membrane receptor genes that included *KLRC1, KLRC3, KLRD1, KIR3DL2, NCR3*, as well as other intra- and extra-cellular effectors that included *SH2D1A, PRF1, GZMB, FASLG, ZAP70, IFNG, CD247*, and *LAT*. Furthermore, the *ZAP70, CD4, IFNG, IL2RB, STAT4, CD247, DLL1, LAT*, and IL12RB2 genes were enriched in COVID-19 mild patients during the acute phase (*Figure 3—source data 1 and 2*). This data indicated that the Th1/Th2 cell differentiation pathway was robust and active during

this phase and likely played an important role in the effective adaptation to dynamic events during the progression of the infection that protected mildly ill patients from experiencing severe symptoms. Interestingly, metabolic pathways involved with hematopoietic cell lineages were enriched in severely ill patients at this matched moment in time with the mildly ill patients (*Figure 3B, D*). Collectively, these observations indicate that coordination between humoral- and cell-mediated immunity were more tightly regulated in mildly ill patients than in severely ill patients.

To confirm the importance of the differentially enriched pathways between mild and severe COVID-19 patients, we focused on analyzing the context of gene–gene interactions (*Figure 4A* and *Figure 4—figure supplement 1*) and changes in their quantitative expression levels overtime graphed as a heatmap (*Figure 4B*). The genes displayed in this KEGG pathway graph represent the upregulated genes (red boxes) in mild patients and their interactions involved in NK cell-mediated cytotoxicity (*Figure 4A*). Interestingly, all these genes showed overtime trajectories with high levels on days 0 and 7 in mild patients. These gene expression levels became roughly equivalent by D28 in both the mild and severe groups (*Figure 4B*). Complementing these observations, we constructed a protein–protein interaction (PPI) network using only upregulated genes during the early phase (days 0 and 7), followed by a clustering process that detected proteins with more significant interactions among the selected genes (*Figure 4C*). Notably, we detected KLRD1, CD247, and IFNG as central nodes of PPI networks. This finding makes sense because these proteins exhibit numerous interactions with other proteins involved in activating or inhibiting NK cell cytotoxicity (e.g., KLRC1, KLRC3, and KIR3DL2), as well as Th1/Th2 cell differentiation (CD4) and cytokine-cytokine receptor interaction (IL5RA and IL2RB). In *Figure 4D*, we show the comparative trajectories of these node genes between both groups of severity. Interestingly, we found a convergence of KLRD1 and CD247 genes on D28, while IFNG remained differentially expressed between patient groups.

Once we identified the trajectories of NK cell hub-genes participating in COVID-19 progression, we asked whether there were any DEGs (adj. p ≤ 0.05 and log$_2$-fold change ≥2.0) obtained from a pairwise comparison of mildly and severely ill patients at days 0 and 7 that would have been left out from the longitudinal analysis. Given that the number of DEGs at each time point is higher when compared to the list of genes exhibiting differential trajectories, we performed a gene ontology (GO) and pathways analysis with the new set list of genes (*Figure 4—figure supplement 2*). The main result showed that NK cell-mediated cytotoxicity was predominant on D7. This finding reinforced the interpretation that there is a dysregulation of innate immunity, as previously suggested in severe patients (*Paludan and Mogensen, 2022*), with an over-representation of neutrophil activation. The results shown in *Figure 4—figure supplement 3* and *Figure 4—source data 1* summarize the pathway and PPI network analysis for these genes on D0 and D7, respectively, and show the predominant enrichment of NK genes. Taken together, these data are consistent with an active and regulated innate NK cytotoxic immune response mounted during the acute phase of infection in mild COVID-19 patients. This observation contrasts with the humoral- and neutrophil-biased response observed in severely ill individuals.

Previous comparisons were done with the assumption that both cohorts were at the peak of their symptoms on D0. However, taking into account the delta at the days of symptoms onset, we also analyzed the pairwise comparison for D7 in mild patients, but this time comparing it to D0 in severe patients. *Figure 4—figure supplement 4* highlights GO terms related to NK cell response and enriched pathways for NK cell cytotoxicity in mild patients. This result supports the idea that the transcriptional program differs between mild and severe individuals with the outstanding contribution of NK cells in mild patients. Otherwise, severe patients exhibit an inflammatory response reflected by the complement and coagulation pathways, which is according to our previous analysis.

## Gene co-expression identifies NK hub-genes linked to the innate and adaptive immune response of mild COVID-19 patients

We identified genes that were coordinately expressed during COVID-19. We developed a weighted gene correlation network to simultaneously analyze all peripheral blood samples collected from patients during the longitudinal protocol and those from heathy donors to identify genes with coordinated expression. By using a differential co-expression approach, we identified 10 modules of co-expressed genes (*Figure 5A*). We then used these networks to correlate each module with available clinical information of the patients by calculating the module significance (MS) for each module–trait

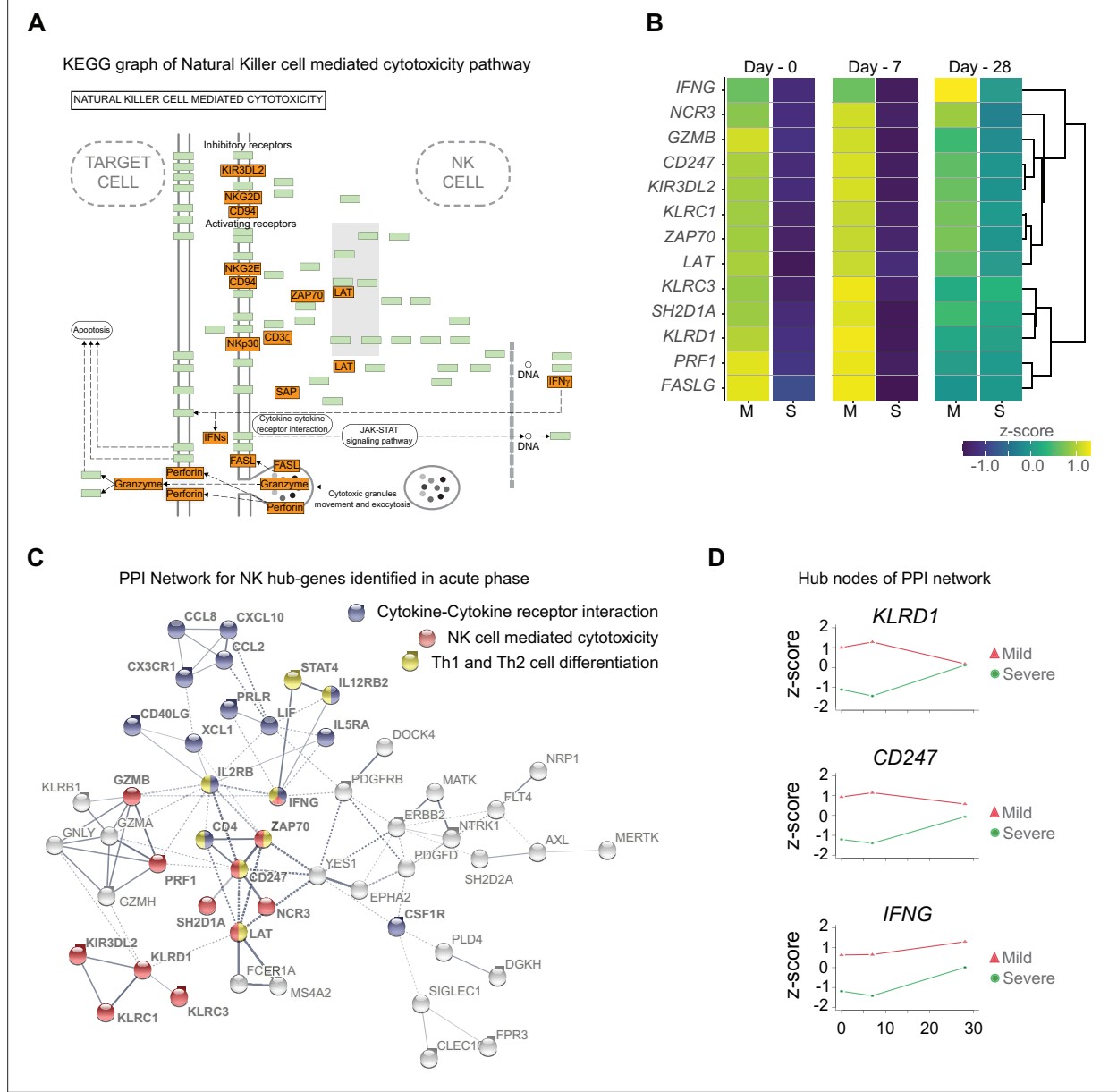

**Figure 4.** Gene function network for natural killer (NK) cell hub-genes with differential expression levels between mild and severe patients during acute phase of COVID-19. (**A**) Kyoto encyclopedia of gene and genomes (KEGG) pathway of NK cell-mediated cytotoxicity represents the set NK cell hub-genes upregulated (red-boxes) in mild versus severe patients. The green boxes correspond to genes without differential expression. (**B**) Heatmap shows the differential expression levels of NK cell hub-genes over time (D0, D7, and D28 after recruitment) separated by mild and severe groups. The expression levels are represented by the z-score of normalized counts. Dendrogram shows the hierarchical clustering of genes. (**C**) Protein–protein interaction (PPI) network for upregulated genes during the acute phase in mild patients. The network corresponds to the principal clusters with more interaction between proteins and highlights the three most represented pathways: Cytokine–cytokine receptor interaction (blue); NK cell-mediated cytotoxicity (red); and Th1 and Th2 cell differentiation (yellow). (**D**) Time course expression levels for the main protein nodes identified in PPI network during the acute phase of COVID-19. The trajectories of these genes are graphed as days after recruitment (D0, D7, and D28) for mild (red triangle) and severe (green circle) groups and their enrichment is represented by the z-score of normalized counts.

The online version of this article includes the following source data and figure supplement(s) for figure 4:

**Source data 1.** Kyoto encyclopedia of gene and genomes (KEGG) graph shows genes with differential expression found in the pairwise comparison (D0 vs D7) from natural killer cell-mediated cytotoxicity pathway for upregulated genes in mild COVID-19 patients at D0 (**A**) and D7 (**B**), from Th1 and Th2 cell differentiation pathway at D0 (**C**) and D7 (**D**), and from cytokine–cytokine receptor interaction pathway at D0 (**E**).

**Figure supplement 1.** Kyoto encyclopedia of gene and genomes (KEGG) graphs show genes differentially expressed over time of the Th1 and Th2 cell differentiation pathway (**A**) and the cytokine–cytokine receptor interaction pathway (**B**).

*Figure 4 continued on next page*

*Figure 4 continued*

**Figure supplement 2.** Gene ontology (GO) and Kyoto encyclopedia of gene and genomes (KEGG) and REACTOME pathway analyses of differentially expressed genes (DEGs) found in the pairwise comparison between D0 and D7 of COVID-19 infection.

**Figure supplement 3.** Protein–protein interaction (PPI) network graphs show the upregulated genes found in the pairwise comparison (D0 and D7) in mild versus severe COVID-19 patients.

**Figure supplement 4.** Gene ontology (GO) and networks of enriched pathway analyses of differentially expressed genes (DEGs) found in the pairwise comparison between D7 of mild patients and D0 of severe patients with COVID-19.

correlation. Not surprisingly, we found that most module eigen genes grouped according to the degree of COVID-19 infection (i.e., mild or severe patients) (*Figure 5B*). Among co-expressed gene modules, we focused on three modules that contained the largest number of genes. These modules correspond to blue (704 genes), brown (508 genes), and turquoise (712 genes). The blue and brown modules, which are correlated positively with mild patients (*Figure 5B*), were enriched with genes related to T cell activation and platelet function, respectively (*Figure 5C*). In contrast, the turquoise module, which was correlated positively with severe COVID-19 patients (*Figure 5B*), was enriched with genes related to neutrophil activation and inflammatory responses (*Figure 5C*).

As shown thus far, the Th1/Th2 cell differentiation pathway was relevant in the immune response of mild COVID-19 patients, and because the blue module is enriched in lymphocyte-based immune response genes, we performed a gene–gene network analysis to determine how the genes from this module might have worked in the context of an adaptive immune response. In this analysis, we found genes belonging to the NK cell-mediated cytotoxicity pathway grouped together with the cytokine–cytokine receptor interaction and Th1/Th2 cell differentiation pathways (*Figure 5D*). Furthermore, these genes were previously identified as differentially expressed in the NK cytolytic pathway, like *KLRD1*, *KLRC3*, and *KLRC1* receptors, as well as *FASLG*, *SH2DB1A/B*, and *LAT*. All of this evidence is consistent with the interpretation that highly interconnected genes from blue module had a functional significance in limiting the progression COVID-19 in mild patients. In the other modules, the brown network (*Figure 5E*) depicts enriched pathways for platelet activation, extracellular matrix–receptor interaction, hematopoietic cell lineage, gap junction, and complement and coagulation cascades which complemented with GO terms is an important focus of interest in COVID19 (*Sb et al., 2024*). Furthermore, the enriched pathways for the turquoise module include leishmaniasis, malaria, osteoclast differentiation, and nitrogen metabolism (*Figure 5F*), some of which are implicated in a neutrophil response as indicated by the GO terms (*Babatunde and Adenuga, 2022*; *Passelli et al., 2021*).

The remaining seven modules were analyzed for GO (*Figure 5—figure supplement 1*) which reveals different aspects related to the immunological response to viral infection. In addition, we searched for enriched pathways and displayed the results in the form of a network graph (*Figure 5—figure supplement 2*) that can be further investigated in future studies.

## Discussion

We systematically analyzed transcriptomic features of PBMCs from COVID-19 patients with mild and severe symptoms at three sequential time points (D0, D7, and D28) during the peak of the symptoms. Our longitudinal analysis revealed key temporal features of immune responses that distinguished mild from severe patients during acute disease. We observed a prominent role of NK cell-mediated cytotoxicity function pathways during COVID-19 progression. These pathways include genes such as *KLRC1*, *KLRC3*, *KLRD1*, *KIR3DL2*, and *NCR3* receptors, as well as other effectors like *SH2D1A*, *PRF1*, *GZMB*, *FASLG*, *ZAP70*, *IFNG*, *CD247*, and *LAT*. Most, if not all, of these genes are implicated in regulatory processes of cytotoxicity and attraction of NK cells as part of the viral infection control mechanism (*Björkström et al., 2022*). Antiviral NK cell cytotoxicity depends on a steady state for survival, basal turn-over and their function maintenance, which are monitored by several checkpoints (*Björkström et al., 2022*; *Masselli et al., 2020*; *Vivier et al., 2008*). We found a dynamic transcriptomic profile of a NK cell gene-hub characterized by higher gene expression levels in individuals with mild disease compared with those with severe symptoms across 0 and 7 days. However, expression levels of these NK cell gene-hubs became more similar between mild and severe patients on D28. In contrast to this orchestrated transcriptional response of dominant NK cells activities, we found an upregulated gene signature consistent with dominant neutrophil activities in our severe cohort even

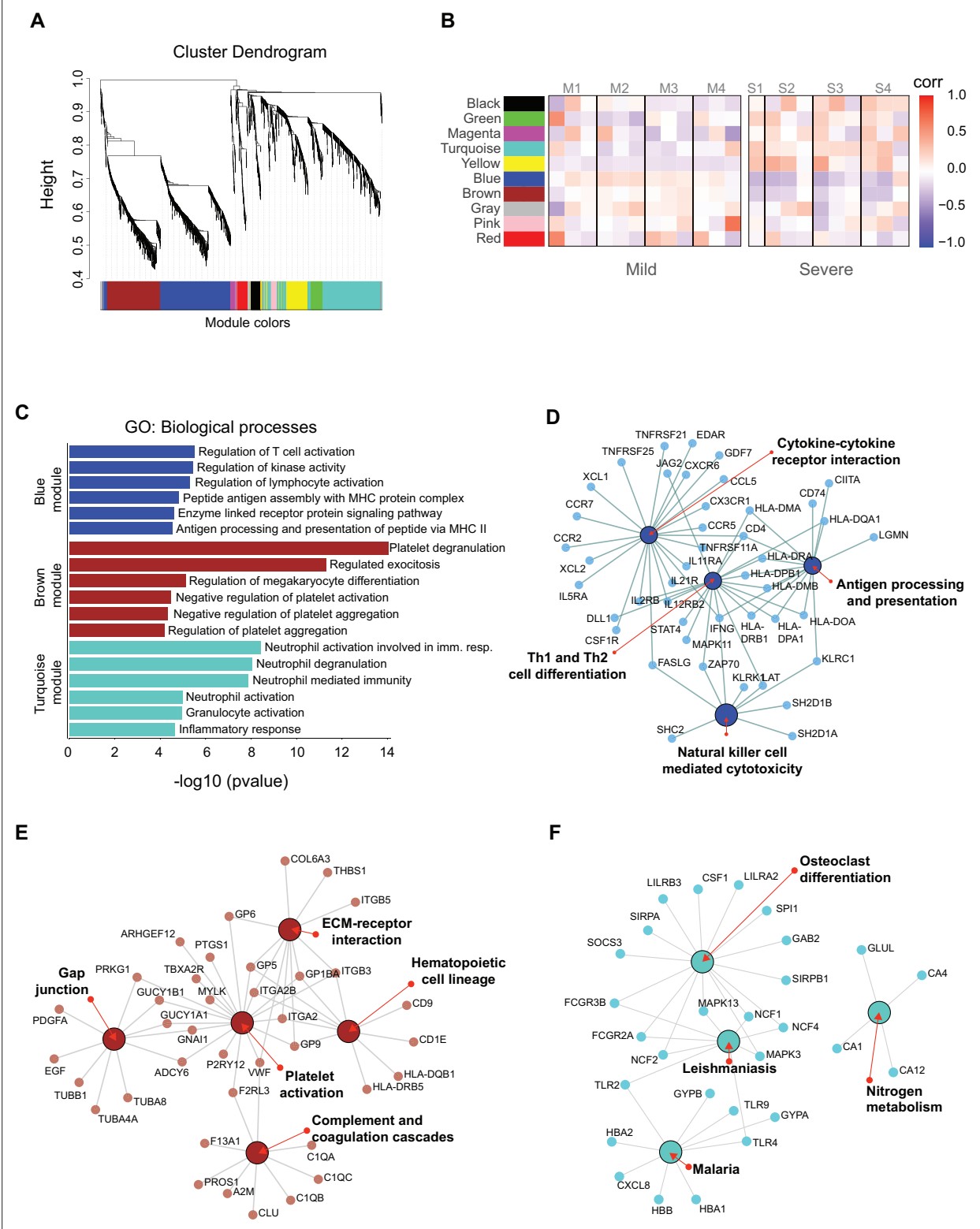

**Figure 5.** Gene co-expression network analysis among the longitudinal transcriptomic profiling. (**A**) Gene hierarchical clustering dendrogram of 10 detected modules based on Topological Overlap Matrices (TOM) measure. The branches and color bands represent the assigned module. (**B**) Module–trait relationships (MTRs) between detected modules and disease severity of COVID-19. MTRs are obtained by calculating Pearson correlation between the traits and the module eigengenes. The red and blue colors indicate strong positive or negative correlations, respectively. Rows represent module eigengene (ME) and columns indicate the disease severity of COVID-19, where correlations from each patient are shown from left to right according to

*Figure 5 continued on next page*

*Figure 5 continued*

their sampling time (D0, D7, and D28). (**C**) Gene ontology (GO) enrichment analysis of genes in the blue, brown, and turquoise modules. The color in the bar graphs refers to the module eigengene (ME). Enrichment results are sorted by −log10(p-value) (higher on top) of each biological process GO term. (**D–F**) Network visualization of enriched pathways based on co-expression genes from blue, brown, and turquoise modules, respectively.

The online version of this article includes the following figure supplement(s) for figure 5:

**Figure supplement 1.** Gene ontology (GO) analysis of the seven smallest modules of co-expression.

**Figure supplement 2.** Network for genes of enriched pathways from the seven smallest modules of co-expression.

after recovery. This finding was previously observed with the concomitant increase of IgG production and complement activation at the earliest phase of disease (*Wang et al., 2020*; *Zhang et al., 2020*; *Zuo et al., 2020*), (*Figure 2*).

In our NK cell gene hub, we recognized activating (*KLRC3*, *NCR3*), and inhibitory (*KLRC1*, *KIR3DL2*) genes of cytotoxicity, as well as regulatory and effector proteins (*KLRD1*, *GZMB*, and *PRF1*). All these genes could be participating in the balancing of a well-coordinated NK cell activity profile. In this sense, the *KRLD1* gene, which encodes the CD94 protein, stands out as an important node interconnecting proteins networks. This node regulates activating (NKG2E from *KLRC3* gene) and inhibitory functions (NKG2A from *KLRC1* gene), and thus modulates NK cell cytotoxicity (*Borrego et al., 2006*). Supporting this role, a previous study demonstrated the importance of CD94:NKG2E heterodimeric receptor in response to the lethal mousepox virus (*Fang et al., 2011*). This node may be relevant for an efficient response against SARS-CoV-2 infection given the high conservation of receptors and ligands between the human and mouse pathways (*Borrego et al., 2006*). *Wauters et al., 2021* found that mild COVID-19 patients displayed an interaction of CD94:NKG2E/HLA-F between their T cells and neutrophils in bronchoalveolar lavage samples. On the other hand, NKG2A is an important inhibitory receptor that interacts with CD94 and together regulate NK cell functions (*Borrego et al., 2005*; *Lee et al., 1998*). Our analysis showed a higher expression of NKG2A in mild than severe patients during the acute phase. Regarding the same receptor, previous research showed that NKG2A was more highly expressed in lymphocytes and NK cells during infection compared with healthy controls (*Zheng et al., 2020*). In parallel, Zheng et al. showed a decrease of expression of NKG2A in recovering patients along with an increase of NK cell number. Collectively, this evidence supports the conclusion that CD94, and its partners, play important roles in regulating both activating and inhibitory checkpoints related to NK cell cytolytic functions. Furthermore, it substantiates the relevance of innate NK cell immune responses in combating SARS-CoV-2, and likely other coronaviruses. This pathway might also be a prominent player in controlling other infectious respiratory virus infections to promote a mild presentation of disease.

Other genes located in the NK cell hub included membrane proteins such as *SH2D1A*, *LAT*, *CD247*, *FASLG*, the enzyme *ZAP70*, and the cytokine *IFNG*. Remarkably, genes encoding *IFNG* and *CD247* were also identified as important nodes within the PPI network during the acute phase. Considering the interactions of these nodes with proteins involved in cytokine–cytokine receptor interactions and Th1/Th2 cell differentiation pathways, it is possible that they coordinately regulated these immune responses with NK cell cytolytic functions. In this context, cytokine-cytokine receptor interaction and Th1/Th2 cell differentiation were well-represented pathways in mild patients during the acute phase highlighting that both innate and adaptive immune responses were active and effective in these patients. Particularly, CD247 (CD3 $\zeta$ ) protein is part of the T cell antigen receptor complex, whose low expression levels have been related to chronic inflammation and decreased T cell activity (*Li et al., 2021a*). In the same line, IFNG protein is a critical player between innate and adaptive immunity after viral infection (*Kang et al., 2018*). Giving support to this connection between innate and adaptive immune responses, it would be expected that adaptive CD8+ T cell cytolytic functions would also be enriched in mild patients due to its important role controlling viral infections (*Prager and Watzl, 2019*; *Uzhachenko and Shanker, 2019*). Interestingly, GO/pathway-based analyses did not detect these functions as a differential player in clinical COVID-19 progression, despite the fact that some genes are shared with NK cytotoxic gene hub (*Uzhachenko and Shanker, 2019*). This observation suggested that Th1/Th2 cell differentiation may be more essential for a successful adaptive response against SARS-CoV-2 than CD8+ T cell cytolytic function in mild patients, at least during the early phase of COVID-19. If this interpretation is credible, then cell-mediated cytolytic activities should rely on the

well-regulated activity of innate NK cell subset as a primary immune response. Taking into account all these data, it is reasonable to interpret that an early fate-compromise toward NK cell activity instead of a neutrophil effector activity may have had an important effect on subsequent processes regulating adaptive immunity. This model favors a robust integration of innate and adaptive immune response during an effective control of COVID-19.

Until now, we have discussed relevant genes involved in immune pathways enriched in mild or severe COVID-19 progression. However, we also decided to look for genes exhibiting coordinated gene expression patterns across all our samples. We found that one module (blue module), which has a strong positive correlation with mild patients, included genes involved with metabolic pathways regulating T cell activation, kinase activity, and antigen presentation. In contrast, the turquoise module, which exhibited a strong positive correlation with severely ill individuals, contained genes associated with neutrophil-related biological processes. This finding was indicative of an opposed early fate of innate immune responses between mild and severe COVID-19 cases. Neutrophil long-term differential enrichment seen across severe cases could be related to other repercussions of SARS-CoV-2 infection, like neutrophil-induced platelet aggregation (*Jevtic and Nazy, 2022*). Consistent with this interpretation, dysfunction of platelets has been associated with abnormal clot formation in severe COVID-19 cases (*Litvinov et al., 2021*). In this sense, our results show that the brown module, which is negatively correlated with severe patients, displays biological processes linked to platelet degranulation activity and negative regulation of aggregation. Hence, these results are consistent with a platelet dysfunction pathology linked to severely ill patients.

We performed a pathway enrichment analysis to understand how the positive correlation of genes in the blue co-expression module related to immune response functions in mildly ill patients. Not surprisingly, the main pathways enriched included Th1/Th2 cell differentiation, cytokine–cytokine receptor interaction, antigen processing, and NK cell-mediated cytotoxicity pathways. Albeit the co-expression analysis included all samples, regardless of the severity of the disease or the longitudinal sampling. This effort revealed transcriptional programs of immune response that were consistent with the profiles detected in mild and severe patients reported in our previous investigations. This evidence supports the idea that the transcriptional regulation of cell-mediated immunity in mildly ill patients is more robust than that observed in patients with severe clinical progression.

In order to identify differences in transcriptional programs associated with mild or severe outcomes, we carefully compared changes in gene expression during the acute phase. This analysis detected a broader list of genes than those found using the longitudinal analysis alone. This accomplishment was resulted from only considering the differences in gene expression between mild and severe groups, independently of quantitative changes in gene expression overtime. These results consistently showed a NK cell hub of genes being differentially expressed in mild patients. Importantly, we found novel DEGs including *KLRK1*, *KIR2DS4*, and *KLRC2*. The gene *KLRK1* codes for NKG2D protein, an activating receptor with critical importance due its interaction with the major histocompatibility complex class I (*Zingoni et al., 2018*). We found that NKG2D is comparatively over-expressed between days 0 and 7 in the mild group. In line with this finding, Varchetta et al. found an increase of circulating NKG2D(−) NK cells using cell cytometry during the acute phase, which was linked to exhaustion in severe COVID-19 patients. Importantly, their sampling times ranged from hours to days after onset symptoms (*Varchetta et al., 2021*). Additionally, the regulatory role of NKG2D in COVID-19 is also supported by *Lee et al., 2022* where their results show that the viral non-structural protein 1 (Nsp1) of SARS-CoV-2 mediates its immune escape by downregulating NKG2D ligands, therefore decreasing NKG2D-dependent NK cytotoxic responsiveness and conferring resistance to infected cells.

Complementary to this scenario of NK cell activating receptors being important in mitigating symptom severity as previously reported (*Gardiner, 2008*), we found that the *KIR2DS4* gene, which belongs to the KIR receptors gene family, was correlated with mild progression of COVID-19. This activating receptor was more highly expressed in mild patients than in severe patients at both 0 and 7 days (4- and 6.8-fold, respectively). In this regard, Bernal et al. found that a low expression of *KIR2DS4* was part of a distinctive immunophenotype in peripheral NK cells that was increased in severe COVID-19 individuals (*Bernal et al., 2021*). On the contrary, *Casado et al., 2022* found an enriched KIR2DS4(+) subset of CD56brightCD16neg peripheral NK cells in hospitalized individuals compared to mild patients, indicating a positive correlation with severity. However, as their mild patient cohort was recruited at a mean of 60 days after diagnosis, a direct comparison with our data

is not precise. Additionally, the absence of severe patients, as well as a lack of longitudinal sampling, were some of the inconsistencies between the study designs of Casado et al. and ours, which may account for these inconsistent observations.

Another NK cell activating receptor (the NKG2C protein) encoded by the *KLRC2* gene was previously implicated as a COVID-19 marker (*Fielding et al., 2022*; *Maucourant et al., 2020*; *Vietzen et al., 2021*). Although we did not find a significant difference in the gene expression of *KLRC2* between mild and severe groups (being excluded by our threshold criteria), we found it to be comparatively lower among severe patients compared to mild patients in both 0 and 7 days (−1.6- and −2.7-fold, respectively). We observed a similar trend in the control patients between 0 and 7 days in the acute phase (−1.6- and −1.7-fold, respectively). Surprisingly, when we compared the quantitative expression of the NKG2C gene in the severe group to the control group on D28, we discovered an inverted pattern with respect to the acute phase in which NKG2C over-passed the levels of controls (2.1-fold). In this context, Maucourant et al., in a scRNA-Seq study performed with bronchio-alveolar lavage from severe COVID-19 patients, showed increased NKG2C levels linked to the adaptive response of NK cells (*Maucourant et al., 2020*). These data additionally reinforce the interpretation that NKG2C is required to mount an effective NK cell response against SARS-CoV-2 infection. *Vietzen et al., 2021* found that a deletion in the NKG2C gene resulted in a significant correlation with severe COVID-19. This evidence supports the idea that the innate and adaptive immune responses are being differentially modulated in severe COVID-19 than mild patients.

Another important observation detected in our comparative analysis included enrichment of the IL-17 pathway in severe patients on D7. Our analysis identified a group of 10 genes (*FOSL1*, *CXCL6*, *CEBPB*, *LCN2*, *TNFAIP3*, *CXCL1*, *CXCL2*, *MMP9*, *S100A9*, and *S100A*) associated with this time point. IL-17 has diverse biological functions that promote protective immunity against many pathogens but also driving inflammatory pathology during autoimmunity. Interleukin-17-driven inflammation is normally controlled by regulatory T cells expressing the anti-inflammatory cytokines IL-10, TGFbeta, and IL-35 (*Pacha et al., 2020*). One explanation may be that an imbalance in T cells and cytokine secretions mediated by IL-17 promote an inflammatory phenotype in patients with severe symptoms. Notably, Th17 cells were elevated on D7 in patients with mild symptoms. These Th17 cells can display plasticity in cytokine production in vivo and can switch from predominantly producing IL-17 to predominantly producing interferon gamma (IFNγ) , thereby resembling Th1 cells (*Lee et al., 2009*). Sequential activation of STAT1 by IFNG and STAT4 by IL-12 drives optimal expression of T-bet (TBX21), a central transcription factor for Th1 programming (*Lee et al., 2009*). Otherwise, the activation of STAT6 by IL-4 upregulates GATA3, which is central to Th2 programming (*Lee et al., 2009*). All the genes related to Th1 activity (*IFNG*, *STAT4*, *TBX21*, and *IL-12*) were upregulated in our cohort of patients exhibiting mild symptoms, which is consistent with this potential regulatory circuit.

Even though the primary criterion for comparing mild with severe was the peak of symptoms in earlier analyses, we also investigate according to the time of the symptoms onset. As a result, we also performed a complement analysis comparing mild-D7 to severe-D0 to identify the enriched pathways that represent this comparison. Notably, the most important GO terms and enriched pathways in mild patients correspond to NK cell cytotoxicity, represented by the same set of genes reported in previous comparisons. This lends significant support to the primary findings of our study, which show that a transcriptional program is differentially regulated in mild patients and is focused on innate immune response associated with NK cell cytotoxicity.

In conclusion, the longitudinal trajectories of gene expression, the differential GO/pathways, and PPI analyses, together with the co-expressed gene–gene correlation network is consistent with the existence of a regulatory transcriptional program linked to an early activation of NK cell cytotoxicity in mild COVID-19 patients. This work establishes the notion that innate immune responses are crucial for the progression of COVID-19 severity, and reinforces the importance of the NK cell cytotoxicity pathway in distinguishing between mild and severe COVID-19 progression. Taken together, these differential responses are complemented by cytokine activities and Th1/Th2 cell differentiation programs indicating a well-regulated crosstalk between innate and adaptive immune responses in the mild COVID-19 progression.

## Materials and methods

### Patient cohort and PBMCs sampling

We recruited a total of eight patients diagnosed to be suffering from COVID-19 who were separated into two groups, one composed of four mild outpatients and another, conformed of four severe hospitalized individuals. Peripheral venous blood samples were obtained by using the venipuncture technique in Vacutainer K2 ethylenediamine-tetraacetic acid (EDTA) tubes (BD, USA) from each patient three times, including two clinical stages (acute phase and convalescence). PBMC was isolated from each fresh heparinized peripheral blood sample, through density gradient centrifugation on Ficoll-Paque Plus (GE Healthcare Life Sciences, USA) by centrifuging at 1600 rpm for 30 min (using minimum acceleration and no deceleration configurations). PBMC-containing fraction was then washed two times with 2 mM EDTA in phosphate-buffered saline (PBS) and stored in RNAlater solution (Sigma, USA) at −20°C until RNA extraction. The detailed clinical features of all patients and the detailed sampling time are shown in *Figure 1* and *Table 1*. All samples were processed in a qualified BSL-2 laboratory and, according to protocols and approval from Institutional Review Boards, CEC-SSLR Ord N°226 and Ord N°399. Written informed consent was received before the participation of each patient.

### RNA extraction, library preparation, and PBMC transcriptome sequencing

Total RNA extraction was performed from PBMC by Diagenode (Belgium). RNA samples were quantified using QubitTM RNA BR Assay Kit (Thermo Fisher Scientific, USA) and secondly checked for integrity using HS RNA Kit (Agilent, USA) on a Fragment analyzer system (Agilent, USA). The library preparation was performed using NEBnext ultraII Directional Kit and sequencing of the samples was performed on an Illumina NovaSeq 6000 instrument producing 150 bp paired-end reads running Control Software 1.7.0.

### Identification of DEGs along COVID-19 progression and between disease severities

Sequencing-quality check was performed using FastQC (*Andrews, 2010*), and low-quality reads were trimmed using Trim_Galore! (*Krueger, 2019*) with `--clip_R1 3` and `--clip_R2 3` options. High-quality reads were aligned to the human reference genome version GRCh38 with STAR v2.6.1a_08–27 (*Zhang et al., 2022*). Transcript counts were generated using featureCounts v1.6.3 (*Bastard et al., 2020*) with default settings. Differential gene expression analysis was performed in two ways using edgeR package v3.36.0 (*Delorey et al., 2021*). Temporally and DEGs and DEGs between mild and severe COVID-19 patients at each sampling time point were identified using the generalized log-linear model option in edgeR. Genes were considered as differentially expressed either with temporal expression differences or disease severity condition using a false discovery rate (Benjamini–Hochberg), an adjusted p-value of <0.05, and an absolute $\log_2$ fold change of 2. Transcript counts (normalized using TMM approach) were used to generate heatmaps for visualization of DEGs using the pheatmap R package. Expression was scaled by row $z$-scores for visualization, taking into account the mild and severe patients to obtain the mean and standard deviation of each group.

### Gene set enrichment analysis

In order to perform gene set enrichment analysis (GSEA) (*Schultze and Aschenbrenner, 2021*), after DEGs analysis only values of $\log_2$ fold change equal to or greater than 1 were considered. Then the GSEA was performed through ClusterProfiler package (v.3.16) (*Yu et al., 2012*) in R, we evaluated the significantly enriched biological process (GO) using gseGO function and pathways from KEGG using gseKEGG function.

### Enrichment analysis of DEGs along COVID-19 progression and between mild and severe patients

GO and pathways enrichment analyses were performed with both upregulated and downregulated genes using Enrichr platform (*Xiong et al., 2020*). Significant GO terms (Biological processes and Molecular functions) and pathways (KEGG and Reactome) were calculated from the adjusted p-value

(*q*-value) using the Benjamini–Hochberg method for correction for multiple hypothesis. Considering a difference of one unit in *z*-score between the two severity groups, for mild patients we analyzed 365, 359, and 101 genes for days 0, 7, and 28, respectively, while for severe patients we analyzed 369, 414, and 136 genes for days 0, 7, and 28, respectively.

## Weighted correlation network analysis

Based on the assumption that DEGs may explain transcriptional differences observed between mild and severe COVID-19 patients. Read counts from DEGs among all samples were selected as a reference set for construction of a weighted gene co-expression networks and modules detection. Co-expressed gene modules were constructed using WGCNA R package v1.71 (*Langfelder and Horvath, 2008*) under a signed networks approach because it provides a better understanding of molecular regulatory mechanisms at the systemic level, facilitating better separation of modules in terms of biological performances. To do so, we removed outliers using the adjacency function and a standardized connectivity score of $<-2.0$; then we used the pickSoftThreshold function to identify the soft thresholding power $\beta$ value, which was subsequently transformed into a Topological Overlap Matrix (TOM). Next, an average linkage hierarchical clustering analysis was performed based on the TOM dissimilarity (1-TOM) and modules were detected through a dynamic hybrid tree cutting algorithm.

After the modules were identified, the module eigengene (ME) was summarized by the first principal component of the module expression levels. Module–trait relationships were estimated using Pearson correlation between MEs and disease severity. To evaluate the correlation strength, we calculated the MS that is defined as the average absolute gene significance of all genes involved in a module.

## PPI network

All genes with differential trajectories over time (#827 genes) were included to construct a PPI network by STRING V11.5 (*Szklarczyk et al., 2021*). After clusterization, we focused our analysis on the principal cluster (one of three) with the highest connectivity between genes and highlighted the genes involved in the most significant pathways according to KEGG.

## Acknowledgements

We especially thank all blood donors who participated in this study. COVID-19 South Chile Group: Renato Ocampo, Christian Esveile, Leonila Ferreira, Johanna Cabrera, Geraro Solís, Marcelo Pozas, Equipo de enfermería Servicio de Medicina Interna Hospital Base Osorno, María Paz Contreras, Equipo de enfermería de la Universidad de Los Lagos Osorno, Catherine Fernández, Camila Rojas, Paulina Lagos, Rocío Mejías, Melissa Canales, Patricio Suazo, Pamela Ángel, Romina Inostroza, Pamela Silva, Felipe Collado, Vanina Cuevas, Rodrigo Oñate, Daniel Salamanca, Javier Briones, Vanessa Villagrán, Diana Bocaz, and Andrés Umanzor. This work was supported by the National Agency of Research and Development (ANID) projects COVID0422 and ANID/ATE220034. We also thank USS-FIN-23-PDOC-02.

## Additional information

### Funding

| Funder | Grant reference number | Author |
|---|---|---|
| National Agency of Research and Development | Project COVID0422 | Raul Riquelme<br>Maria Luisa Rioseco<br>Mario Calvo<br>Jose Luis Garrido<br>Maria Ines Barria |
| National Agency of Research and Development | ATE220034 | Felipe Aguilera<br>Ian Burbulis<br>Jose Luis Garrido<br>Maria Ines Barria |

| Funder | Grant reference number | Author |
|---|---|---|
| Universidad San Sebastian | USS-FIN-23-PDOC-02 | Camila Kossack |

The funders had no role in study design, data collection, and interpretation, or the decision to submit the work for publication.

## Author contributions

Matias A Medina, Conceptualization, Data curation, Formal analysis, Investigation, Visualization, Methodology, Writing - original draft, Writing – review and editing; Francisco Fuentes-Villalobos, Investigation, Methodology, Writing - original draft; Claudio Quevedo, Data curation, Formal analysis, Investigation, Visualization; Felipe Aguilera, Data curation, Formal analysis, Supervision, Funding acquisition, Investigation, Writing – review and editing; Raul Riquelme, Maria Luisa Rioseco, Yazmin Pinos, Mario Calvo, Resources, Supervision; Sebastian Barria, Resources; Ian Burbulis, Funding acquisition, Writing – review and editing; Camila Kossack, Funding acquisition, Investigation, Writing – review and editing; Raymond A Alvarez, Conceptualization; Jose Luis Garrido, Conceptualization, Supervision, Funding acquisition, Investigation, Methodology, Writing – review and editing; Maria Ines Barria, Conceptualization, Supervision, Funding acquisition, Investigation, Methodology, Project administration, Writing – review and editing

## Author ORCIDs

Matias A Medina ⓘ https://orcid.org/0000-0002-1905-6771
Francisco Fuentes-Villalobos ⓘ https://orcid.org/0000-0003-3120-0198
Felipe Aguilera ⓘ https://orcid.org/0000-0003-3235-931X
Camila Kossack ⓘ http://orcid.org/0000-0002-3036-8813
Maria Ines Barria ⓘ https://orcid.org/0000-0001-6225-5971

## Ethics

Human blood samples were collected after signed informed consent that was obtained in accordance with protocols and approval from the Institutional Review Board (Servicio de Salud Los Ríos), CEC-SSLR Ord N 226 and Ord N 399.

Reviewer #1 (Public Review): https://doi.org/10.7554/eLife.94242.3.sa1
Reviewer #2 (Public Review): https://doi.org/10.7554/eLife.94242.3.sa2
Author response https://doi.org/10.7554/eLife.94242.3.sa3

---

# Additional files

## Supplementary files

• MDAR checklist

## Data availability

Sequencing data have been deposited in BioProject ID PRJNA1157471.

The following dataset was generated:

| Author(s) | Year | Dataset title | Dataset URL | Database and Identifier |
|---|---|---|---|---|
| Medina MA, Fuentes-Villalobos F, Quevedo C, Aguilera F, Riquelme R, Rioseco ML, Barria S, Calvo M, Pinos Y, Burbulis I, Kossack C, Alvarez RA, Garrido JL, Barria MI | 2024 | Longitudinal transcriptional changes reveal genes from the natural killer cell-mediated cytotoxicity pathway as critical players underlying COVID-19 progression | https://www.ncbi.nlm.nih.gov/bioproject/PRJNA1157471 | NCBI BioProject, PRJNA1157471 |

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
