## [Editor Report · eLife assessment]

This **valuable** paper compares blood gene signature responses between small cohorts of individuals with mild and severe COVID-19. The authors provide **solid** evidence for distinct transcriptional profiles during early COVID-19 infections that may be predictive of severity, within the limitations of studying human patients displaying heterogeneity in infection timelines and limited cohort size.

---

## [Referee Report · Reviewer #1 (Public Review)]

Summary:

Medina et al, 2023 investigated the peripheral blood transcriptional responses in patients with diversifying disease outcomes. The authors characterized the blood transcriptome of four non-hospitalized individuals presenting mild disease and four patients hospitalized with severe disease. These individuals were observed longitudinally at three timepoints (0-, 7-, and 28-days post recruitment), and distinct transcriptional responses were observed between severe hospitalized patients and mild non-hospitalized individuals, especially during 0- and 7-day collection timepoints. Particularly, the authors found that increased expression of genes associated with NK cell cytotoxicity is associated with mild outcomes. Additional co-regulated gene network analyses positively correlates T cell activity with mild disease and neutrophil degranulation with severe disease.

Strengths:

The longitudinal measurements in individual participants at consistent collection intervals can offer an added dimension to the dataset that involves temporal trajectories of genes associated with disease outcomes and is a key strength of the study. The use of co-expressed gene networks specific to the cohort to complement enrichment results obtained from pre-determined gene sets can offer valuable insights into new associations/networks associated with disease progression and warrants further analyses on the biological functions enriched within these co-expressed network modules.

Weaknesses:

There is a large difference in the infection timeline (onset of symptom to recruitment) between mild and severe patient cohort. As immune responses during early infection can be highly dynamic, the differences in infection timeline may bias transcriptional signatures observed between the groups. The study is also limited by a small cohort size.

Comments on revised version:

The authors have addressed the specific concerns brought forth by the reviewers.

---

## [Referee Report · Reviewer #2 (Public Review)]

In their manuscript, Medina and colleagues investigate transcriptional differences between mild and severe SARS-CoV-2 infections. Their analyses are very comprehensive incorporating a multitude of bioinformatics tools ranging from PCA plots, GSEA and DEG analysis, protein-protein interaction network, and weighted correlation network analyses. They conclude that in mild COVID-19 infection NK cell functionality is compromised and this is connected to cytokine interactions and Th1/Th2 cell differentiation pathways cross-talk, bridging the innate and the adaptive arms of the immune system. The authors successfully recruited participants with both mild and severe COVID-19 between November 2020 to May 2021. The analyzed cohort is gender and acceptably age-matched and the results reported are promising. Signatures associated with NK cell cytotoxicity in mild and neutrophil functions in the severe group during acute infection are the chief findings reported in this manuscript.

Comments on revised version:

The authors responded appropriately to the previous review critiques.

---

## [Author Response]

The following is the authors’ response to the original reviews.

**Reviewer #1 (Recommendations For The Authors):**
(1) Due to the significant difference between the infection timeline of mild (1 day post symptom onset) and severe (10 days post symptom onset) cohort at enrollment, an informative analysis to consider is to compare timepoint 2 from the mild cohort to timepoint 1 from the severe cohort.

In agreement with what the reviewer noted on his comment, to be more helpful we completed the analysis comparing timepoint 2 from the mild cohort to timepoint 1 from severe cohort, which is now included as Figure 4-figure supplement 5. The new text added is on pages 13-14, lines 346-355 explaining this analysis. We also included a paragraph in the discussion on page 22, lines 595-604. We have resolved to show this comparison to enforce the main observation related to Natural Killer Cytotoxicity pathways enriched in all analyses of this work.

(2) Alternatively, as this information is available, the authors may group the samples based on the individual's infection timeline as opposed to the recruitment timeline.

Patients in both groups were enrolled at the peak of their symptoms. According to this criterion, we grouped the patients to generate more significant results. Since these infections occurred naturally, we have no accurate information regarding the infection timing of patients. However, if the samples were grouped in order of individual infection timeline, the analysis would be statistically weak to make conclusions about the course of COVID-19, as disease progression would not be coordinated. Our grouping approach provided us a good confidence range, despite the tiny population evaluated.

(3) The authors selected three co-regulated network modules based on the size of module membership genes, selecting the three modules containing the largest gene membership. Small co-regulated networks can also offer important biological insights into specific molecular machinery associated with disease outcomes.

Figure 5 was updated including two more networks (besides blue), for brown and turquoise modules (5E and 5F). This new information allowed us to understand deeply the three larger modules with the most significant results, due to the number of genes they included (blue: 704, brown: 508, and turquoise: 712). The new text describing this analysis is included in page 15 lines 388-396. The remaining 7 modules were also analyzed, and the Gene Ontology/Pathways enrichment were included in 2 new supplemental figures (Figure 5 - figure supplement 1 and 2). The new text describing this analysis is included on page 15, lines 397-401.

(4) An alternative selection criterion that can inform biological associations between module genes and disease severity is the strength of the correlation coefficients. It seems from Figure 5B, that yellow, turquoise, and green modules have a moderate positive correlation with severe patients, while brown, blue, and gray modules show a slight positive correlation with mild outpatients. A recommendation for the authors is to consider revising Figure 5C to include the enrichment of these additional modules and include these modules in the interpretation of the results.

The correlations between cohorts and the modules (blue, brown and turquoise) are clearly identified for severe or mild patients. However, for several smaller modules, correlations are heterogenous for different patients of the cohorts, making it hard to gain a clear conclusion related to severity groups. In this sense, the 7 modules were analyzed as is indicated in the previous response number #3, and the results offer an idea of the different transcriptional programs present at different patients in different stages of disease. However, the small number of genes in some modules brings weak results of GO and enriched pathways, making it difficult to interpretation. The text describing this figure is included in page 15 lines 397-401. Also, the network analyses for brown and turquoise modules were included in figure 5 as 5E-F and the text detailing these figures was included on page 15 lines 388-396.

(5) In Figures 3E and 3F, the authors present enrichment analyses of differentially expressed genes from day 28. However, earlier in the results (lines 226-228), the authors reported no differentially expressed genes observed between the mild and severe participant cohort at this time point. Can the authors clarify which comparison was performed to obtain the list of differentially expressed genes used in the enrichment analyses in Figures 3E and 3F?

The discrepancy in this case stems from separate criteria employed for comparison in each case. At the pairwise comparison, DEGs list is different from the longitudinal comparison mentioned afterwards, as for this later analysis we selected only the genes with different trajectories throughout the study (Figure 3). To clarify this point, we included a new paragraph on page 11, lines 278-285.

Original:

“We detected 828 genes that exhibited temporal and quantitative expression level differences during the progression of disease. We discovered additional biological processes and KEGG pathways that were differentially enriched during the COVID-19 progression in mild and severe patients (Figure 3) using the Enrichr platform (G. Chen et al., 2020)”

Changed to:

“To do so, we first identified genes that were differentially expressed between severity groups, and second, we chose only those that also showed changes in their trajectories across sampling times. In doing so, we found 828 genes that exhibited temporal differences in expression level during disease progression. Then using the Enrichr platform (G. Chen et al., 2020), we discovered additional biological processes and KEGG pathways that were differentially enriched during the COVID-19 progression in mild and severe patients (Figure 3).”

(6) Additionally, the authors refer to specific enriched genes in Figure 3 (lines 298-302), but Figure 3 only displays the enriched terms. Can the authors include the results from the enrichment analysis that include gene membership for each enriched term in the supplement?

Certainly, there is no figure or table in the initial version that includes the gene list for this analysis. We have now included a supplement table 1 and 2 that details each pathway, along with its gene list.

(7) In line 104, can the authors clarify the parameters used to define well-matched samples?

Based on the observations made by the reviewers, we decided to change the wording to make it more obvious about the message of this paper. The update was included on page 5, line as follows:

Original:

“Here, we designed a longitudinal investigation using well-matched samples to study how changes in gene expression in distinct immune effector cells changed during the earliest time points after diagnosis and during progression of clinical disease”,

Changed to:

“Here, we designed a longitudinal comparison between mild and severe patients, choosing the appropriate samples according to the clinical progression and the unbiased gene expression profile”

(8) In lines 113-116, can the authors clarify how their approach mitigates noise/potential biases and very briefly, describe what the nature of noise/biases could be?

The main goal of this paragraph is to show that, while there are several pathways with statistical significance in our analyses, the focus was on NK cell cytotoxicity because this molecular pathway showed bridges between other relevant immune responses; thus, the pathways chosen to respond to its intricated transcriptional program instead of a biased interest. The text was edited and included on page 6, line 111-131 as follows:

Original:

“We used a pairwise comparison of gene expression, gene set enrichment, and weight-correlated gene network analyses to detect differential expression of genes involved with the cytotoxic signaling pathway of Natural Killer (NK) cells in mild verses severe progression of disease. We promoted a broad and integrated point of view throughout the transcriptomic analysis of functional pathways to mitigate noise and potential biases (Bastard et al., 2020; Delorey et al., 2021; Schultze & Aschenbrenner, 2021; S. Zhang et al., 2022). We found close connectivity between NK signaling pathway genes and those of cytokine-cytokine receptor signaling pathways, along with Th1/Th2 cell differentiation genes, as part of the transcriptional circuit executed preferentially among mildly ill patients. Our results detected transcriptional circuits engaging multiple regulatory checkpoints. These findings indicated that the innate NK signaling pathway (cell cytotoxic activity) is beneficial, perhaps a critically-necessary activity needed to effectively eradicate coronavirus. We interpreted that an adaptive immune response that included early cell-mediated immunity was important for reducing disease severity in mild patients. This balance between humoral- and cell-mediated immunity appeared to be less robust in patients presenting with severe COVID-19. These results detected components of the immune response that were significantly associated with the differences in symptom severity observed between mild and severely ill COVID-19 patients.”

Changed to:

“Briefly, to gain more insights into our findings and complement their functional context, we used a pairwise comparison of gene expression, gene set enrichment, and weight-correlated gene network analyses. By doing so, we identified pathways of genes involved with the NK cell cytotoxicity enriched in mild patients when compared to severe. Besides focusing on a particular molecular pathway, we investigated the interactions to better comprehend the underlying phenomena of a successful immune response, contributing to an integrated point of view throughout the transcriptomic analyses of functional pathways to mitigate potential biases attributed to focusing the study on a single pathway. In this regard, we revealed that the NK signaling pathway was intricately related to other transcriptional circuits, such as those governing Th1/Th2 cell differentiation and cytokine-cytokine receptor signaling pathways. These interactions highlight the importance of these pathways as bridges between the innate and adaptive immune responses throughout the disease, implying that the innate NK signaling pathway (cell cytotoxic activity) is beneficial, and possibly a critical activity required to effectively eradicate coronavirus. We also concluded that an adaptive immune response including early cell-mediated immunity was significant in lowering disease severity. The link between the primary innate NK cell activity and the transcriptional priming of adaptive Th1 and Th2 cell responses appears to be more robust in mild patients than in severe.”

(9) In line 120, can the authors clarify which regulatory checkpoints were being referred to?

The concept of “checkpoint” was changed to “bridges” (line 124), because offers a clearer idea about the molecular interaction displayed across the different enriched pathways described in our study. In this sense, the bridges show the connection between innate immune response by NK cell and the adaptive immune response by Th1/Th2 cells

(10) In lines 125-126, can the authors refer to specific results to support this observation?

Lines 111 to 129 summarize the results of the analysis that support the aforementioned phrase. However, the original sentence referred was modified for better comprehension on page 6, lines 129-131 as follows:

Original:

“This balance between humoral- and cell-mediated immunity appeared to be less robust in patients presenting with severe COVID-19”

Changed to:

“The link between the primary innate NK cell activity and the transcriptional priming of adaptive Th1 and Th2 cell responses appears to be more robust in mild patients than in severe.”

(11) In lines 184-185, can the authors clarify what the term "mixed" specifically refers to?

The original text was modified for better comprehension on page 8, lines 177-179 as follows:

Original:

“Interestingly, on day-28, when the majority of patients had recovered, samples from severely ill patients were still mixed compared to those with mild symptoms.”

Changed to:

“Interestingly, on day-28, when the majority of patients had recovered, samples from severely ill patients were pooled together with those mild patients who had already recovered”.

(12) In line 286, can the authors clarify how quantitative expression level differences are distinct from temporal expression level differences?

Despite the differences in the enrollment time between mild and severe cohorts, it was made precisely during COVID-19 symptoms peaks, as illustrated in figure 1B. Also supporting this criterion, the longitudinal analysis outlined in figure 3 was performed taking into account the changes in gene expression trajectories along all sampling times. This point has significance because the results obtained from it exposed several transcriptional programs that were dynamically executed along disease progression, even independently of the pairwise comparison approaches carried out previously.

(13) In Figure 1C, there seemed to be two data points associated with "M1 0 days" and "M4 28 days" with distinct PC projections. Could these samples be mislabeled?

The figure was revised and completed. The hexagon symbol for day-28 was changed for a star symbol. The “M1 0 days” and “M4 28 days” samples were labeled correctly. See below figure 1C with changes as follows:

(14) In Figure 1D caption: could authors clarify if the ranking of 100 genes was based on the log2FC or adjusted p-values?

The criteria considered was Fold Change ≥ 2 and the FDR ≤ 0.05 which is included in the methodology on page 23, lines 657-660

(15) In Figure 4D, can the authors include the expression z score for the healthy participants?

We could include this information, but we consider that it would not help for the understanding of this figure because in this way we put the focus on the differential trajectories between mild and severe patients. Also, DEGs from mild and severe cohorts from this analysis or any other in this work were obtained relatively to healthy donors.

(16) Related to this, can the authors clarify if the expression z scores were computed using the mean and standard deviations of all samples within the study or relative to a specific participant cohort?

The z-score was used considering the mild and severe patients to calculate mean and then the standard deviation of each group. A new paragraph was included in material and methods on page 24, lines 662-664.

(17) In Figure 5B, can the authors include column annotations for participants and sampling time points?

The figure 5B was updated and completed with the suggested information.

(18) In Figure 1 - Figure Supplement 2, can the authors include the volcano plot from the pairwise comparison for day 28 showing no differentially expressed genes between mild and severe participants as reported in the results (lines 226-228)?

The third volcano plot for day 28 was included in the updated figure 1 supplement 2.

**Reviewer #2 (Recommendations For The Authors):**
The manuscript is generally very well-constructed and well-written. However, the following are the major concerns mostly regarding the study design and participant selection.(1) The authors have used enrolment day as D0 which is not reflective of the immune response timeline. Especially when the designated 'D0' for the severe group is 10.0 + 1.8 days post symptom (DPS) onset while the 'D0' for the mild group is 1.2 + 1.3 DPS. In the context of an acute infection discussed herewith, this difference is critical.As tempting as it is to conduct longitudinal studies on COVID-19, the authors might do better focusing on specific acute time points (within 10 days post-symptom onset) and convalescent time points (beyond 28 days post-symptom). A better comparison would be D0 severe with D7 mild (aligning the DPS to be between 7-10 days in both groups).

Despite the differences in the enrolment time between mild and severe cohorts, it was made precisely during COVID-19 symptoms peaks, as illustrated in figure 1B. Also supporting this criterion, the longitudinal analysis outlined in figure 3 was performed taking into account the changes in gene expression trajectories along all sampling times. This point has significance because the results obtained from it exposed several transcriptional programs that were dynamically executed along disease progression, even independently of the pairwise comparison approaches carried out previously. Likewise, we agree with the observation of the reviewer, because as we mentioned in the article, it is difficult to properly compare disease progression between naturally infected patients. So, to better support our findings, we complemented them throughout a pairwise comparison between day-7 samples from mild and day-0 samples from severely ill individuals, finding GO terms and enriched pathways related to NK cell function across the mild cohort, as seen in Figure 4-figure supplement 5. This result enforced the main findings gained from the different analyses carried out in this work, highlighting the relevance of the innate immune response of Natural Killer cells, which correlated with a mild progression of disease. The new paragraph describing this analysis was included in pages 13-14, lines 346-355. We also included a paragraph in the discussion on page 22, lines 595-604.

(2) Though there are four participants within each group, one of the participants with severe infection (S1) only has the D0 time point which probably undermines the statistical significance of the results.

This is an accurate observation, as the statistical weight will allow the deeper alterations to be evaluated while the more subtle ones will most likely be excluded from this study. In our analyses, we focused on variations with high statistical significance, which led to the discovery of a distinct Natural Killer response between mild and severe cohorts.

(3) The authors should also account for any medications administered to the severe group in the ICU before enrolment in the study -immune-dampening drugs or steroids which may alter neutrophil recruitment or other immune functions.

Only one severe patient received medication both prior to and during the COVID-19 disease. Even though several medications were administered to this patient, their effects have not been found to increase the neutrophil response.

(4) What was the viral load status at the different time points analyzed - how does this relate to the immune and clinical findings?

In this recruitment the viral load status was not measured.

(5) Was any complete blood count or basic immune phenotyping conducted on these samples? Important to know the various cell frequencies in the PBMC mix sent for sequencing to account for contamination of lymphocytes with RBCs/monocytes/neutrophils as well as any lymphopenia.

This measurement was not done for these samples. However, our protocol of PBMC purification has been tested before and showed small quantities of red blood cell contamination in the process. Furthermore, in all analysis of Gene Ontology or Enriched Pathways, there is not any related to red blood cell genes that could generate noise in the interpretation of our results.

(6) The neutrophil/lymphocyte ratio is already skewed during SARS-CoV-2 infection - which could be the reason for higher readings in severe participants? - speculate?

Effectively, the ratio in several cell types is changed during SARS-CoV-2 infection. However, despite this noise in the proportion of immune cells, different functions in our study are more represented in cells with less count as Natural Killer cells. The modules of co-expression analysis support the notion that despite the number of cells being in different proportions, a transcriptional program is being executed differentially in the cohorts.

(7) CD247/ZAP70 also influences the CD16-mediated NK cell ADCC activity which the authors can add to the innate-adaptive bridging section.

NK CD16a is more highly expressed in NK cells. The circuit involving CD247/ZAP70 and CD16 could explain the cytotoxicity of these cells and how they contribute to the establishment of a response to fight the viral infection of SARS-CoV-2. In our study, CD16a (FcgammaRIIIa) expression was similar in both mild and severe cohorts. Because our methodology only counts transcriptional changes, genes that did not change were excluded from our discussion. However, our group's research focuses on this node or bridge between innate and adaptive immune responses, with a particular emphasis on fc-antibodies functions, being a topic of interest for future research.

(8) Some of the figures lacked clarity making it difficult to review. (Eg. Fig 4 A, Fig 4 - supplement 1 A&B, Fig 5).

Figure 4A was redesigned, Figure 4-figure supplement 1 was presented in a full page for better resolution.

Specific Comments:(1) Consider changing "covid-19" in the title of the manuscript to "COVID-19"

Probably the journal platform changes the letters. The original title is in capital letters according to the observation. In the clinical table “COVID-19” was changed to capital letters.

(2) Page 2: Line 24 - Consider revising this line. Not sure what the authors mean by 'early compromise'

The paragraph was revised and rewritten.

Original:

“Mild COVID-19 patients presented an early compromise with NK cell function, whereas severe patients do so with neutrophil function”

Changed to:

”Mild COVID-19 patients displayed an early transcriptional commitment with NK cell function, whereas severe patients do so with neutrophil function”

(3) Page 4: Lines 57 & 58 - Verify the reference. The paper referenced was published in 2016 and is in regard to SARS-CoV, MERS-CoV, and enterovirus D68.

Effectively, this reference was appropriate for drawing parallels with other respiratory viruses. Due to the emphasis on SARS-CoV-2, the paragraph has been strengthened with two additional references: Shen 2023, and Wauters 2022.

(4) Page 10: Lines 229 - 234 - Consider referring to the appropriate figure (i.e., Figure Supplement 2 A or B). The figure associated with D28 DEGs (Volcano plot) is missing in the supplementary. Erroneously referred here as Figure 1C which is a PCA plot?

The original text was changed because the figure referenced was correct but misunderstood. The final sentence is on page 9, lines 220-223.

(5) Page 10: Line 224 - Change the sentence to " We found upregulated.." instead of " We found regulated..".

The text was edited in accordance with this recommendation, which is currently found in line 232.

(6) Page 13: Line 326 - Figure 4A referenced here is not clear - unable to review.

Figure 4A was updated for a better resolution and included in the manuscript.

(7) Page 15: Line 398 - Consider rewording "after diagnosis" since the days here are "after enrolment".

This recommendation was considered and the text was rewritten on page 15, lines 404-406:

Original:

“We systematically analyzed transcriptomic features of PBMCs from COVID-19 patients with mild and severe symptoms at three sequential time-points (D0, D7, and D28) after diagnosis”

Changed to:

“We systematically analyzed transcriptomic features of PBMCs from COVID-19 patients with mild and severe symptoms at three sequential time-points (D0, D7, and D28) during the peak of the symptoms”

(8) Page 17: Move text from the next page to eliminate blank space.

Resolved

(9) Page 32: Figure 1C - Consider changing the symbol for D28 since it looks very similar to the D0 symbol. Use the colors consistently instead of different shades for each group.

The hexagon symbol was changed by a star symbol for D28 in figure 1C. In this figure each color indicates the three different groups, and the transparent color was used to differentiate the symbols when are close together.

(10) Page 36: Figure 4A - Unable to review.

This figure was resized for better resolution.

(11) Page 42-49: Consider relabeling and renumbering the Supplementary figures for consistency and reference the modified numbers in the appropriate location in the main text.

The supplementary figures were relabeling for consistency and better understanding.

(12) Pages 44 & 48: Unable to review the figures.

The figures indicated were resized for better resolution.

Examples of consistency review:(1) Use of D0,D7 / D-0, D-7 throughout the manuscript

The selected format for the final version of the manuscript is D0, D7, and D28.

(2) Reporting the source of reagents consistently (Name, Place, Country, Catalog number)

The source reagents were reformatted for consistency in lines 626-628-632-642.